# The Role of The Internal Auditor in Strengthening the Governance of Economic Organizations Using the Three Lines of Defense Model

Omar Ikbal Tawfik [1,*], Omar Durrah [2] and Karima Ali Aljawhar [3]

[1] Department of Accounting, Dhofar University, Dhofar, Salalah 211, Oman
[2] Department of Management, Dhofar University, Dhofar, Salalah 211, Oman; odurrah@du.edu.om
[3] Department of Accounting, Mustansiriyah University, Baghdad 14022, Iraq; karima225@yahoo.com
[*] Correspondence: otawfik@du.edu.om

**Abstract:** Purpose: This paper aims to investigate the impact of the three lines of defense (TLOD) in strengthening corporate governance in industrial companies in the Sultanate of Oman. Methodology: A questionnaire was used to collect data from industrial companies in the Sultanate of Oman. A total of 300 questionnaires were distributed; for the 159 valid questionnaires used for analysis, PLS-SEM was used in the data analysis. Results: The results showed a significant impact of the three variables (commitment of operational management to legal, regulatory, and ethical requirements; risk management, compliance, and quality functions; and the role of assertive internal auditing according to the third line of defense model) in strengthening corporate governance. Practical implications: The study indicates that the TLOD model plays a more decisive role in determining the strengthening of corporate governance, and therefore, the results of the study can help industrial companies to understand the role of the TLOD model in strengthening control procedures, risk management, and governance. Originality/value: The study constitutes a management strategy that assists organizations in diagnosing the degree of corporate compliance with the TLOD and identifying weaknesses in their procedures.

**Keywords:** three lines of defense model; corporate governance; industrial companies; Sultanate of Oman

## 1. Introduction

The separation of ownership and management leads to management sometimes carrying out activities that are harmful to the interests of shareholders, which leads to an agency problem between management and shareholders. To solve the agency problem, good corporate governance (CG) is required to regulate the relationship between the actors of the company and to define the rights and responsibilities of each party (Brickley and Zimmerman 2010). The CG system provides an infrastructure that contributes to reducing the cost of capital, achieving a high level of performance, enhancing corporate competitiveness, and ultimately creating sustainable wealth for shareholders (Agyemang et al. 2013).

The issue of CG is of great practical importance in both developed and developing economies. Governance plays a major role in the management of organizations in all countries regardless of governance structure, ownership structure, level of country, or company (Davies and Schlitzer 2008). Castrillo et al. (2010) asserts that there is no integrated model (ideal or standard) for CG that can be applied to all countries and all companies. Several studies suggest that effective CG depends on "Compatibility of organizational and environmental characteristics" (Aguilera et al. 2008; Aguilera and Desender 2012). As a result, governance functions have evolved to counteract fraud and manipulation. This evolution included classic control systems such as the internal control system (ICS), risk management system (RMS), and internal auditing (see Gramling et al. 2004; Behrend and Eulerich 2019).

To meet these risks and challenges, the Institute of Internal Auditors (IIA) has proclaimed the TLOD model for risk control since 2011. Several studies have confirmed that the three lines of defense model is an effective model that can be used in risk management (Decaux and Sarens 2015; EY 2013; The Institute of Internal Auditors (IIA) 2013; KPMG 2012; PWC 2017). It has been considered the best practice for companies and a regulatory model required by banking regulators such as the Basel Committee on Banking Supervision to respond to inefficient risk management during financial crises (Minto and Arndorfer 2015; Bantleon et al. 2021). According to the paper released by The Institute of Internal Auditors (IIA) (2013), The TLOD provides a simple and effective way to enhance risk management and control communication by clarifying essential roles and duties.

The first line of defense (LOD) expresses that the executive departments in the organization are operational or service departments. The required first LOD is to create self-monitoring mechanisms to follow up on the daily operational work. As for the second LOD, this refers to departments assisting in setting control mechanisms for the first LOD, and then examining and measuring the achieved and unrealized performance in the first LOD and submitting reports to the executive management in the organization, such as the executive director or the undersecretaries of the ministry.

In 2020, the IIA renewed the model to reflect changes in concepts and issues related to risk management and governance. The new project included a comprehensive review of approaches to control worldwide, an analysis of how the legacy paradigm is being incorporated into rule and regulation, and a compilation of internationally recognized experts and opinion leaders' feedback. In the recent update of the three lines model, the new model was built around the idea that good governance encourages goal achievement, and that thoughtful risk management is one of the actions. This requires that governance gives the organization scope to pursue goals that involve a certain degree of risk, such as mergers, product development, new sales strategies, or something else.

This study adds value to the literature on the role of internal auditing and CG by applying the TLOD model and examining its impact on CG. The research provides powerful practical insights into the adoption and use of the TLOD in developing countries such as the Sultanate of Oman. The research aims to study the role of the TLOD in strengthening CG in industrial companies in the Sultanate of Oman. The main question of the research is: Does the three lines model contribute to strengthening CG in the economic units of the same research? What are the lines most committed by the industrial companies in the Sultanate of Oman? To achieve the goal of the research, a questionnaire was designed based on the TLOD model and previous studies. The results of the study showed that there is a positive relationship between the TLOD (operational management compliance with legal, regulatory, and ethical requirements, risk management, compliance, quality functions, and assurance internal audit) and strengthening governance.

The results of this research provide valuable recommendations to policymakers and researchers and will enable them to learn more about the key features of the successful adoption of the three lines of defense. The study makes several contributions: first, encouraging industrial companies to benefit from the advantages of the system of applying the three lines of defense; second, encouraging the executive authorities to implement risk management practices and continue to improve them; and third, the study contributes to assisting internal auditing bodies in providing advice to management on the adequacy and effectiveness of governance and risk management procedures.

## 2. Theoretical Framework and Hypothesis Development

### 2.1. Three Lines of Defense Model

The TLOD model provides a simple and effective way to enhance communication on risk management and control by clarifying essential roles and duties. The TLOD model has become a reliable and effective tool in a wide range of industries. The model includes three basic lines.

### 2.1.1. First Line: Operational Management

The first LOD relates to the risk-causing functions, which are the operational management that identify and manage risks. The first LOD management is required to create internal self-control mechanisms to follow up on the daily work and take appropriate corrective actions to address deviations, in addition to verifying that the control procedures are working effectively. One of the simplest control mechanisms is performance indicators that show what is being achieved regularly (Leech and Hanlon 2016). Operational management is responsible for maintaining effective internal controls and implementing risk control procedures daily. Following governance frameworks and legislation such as COSO, SOX, and King III, the board of directors is primarily responsible for establishing and maintaining an appropriate and effective internal control structure and verifying the existence of appropriate control mechanisms and effectively (Moeller 2013). Principle 2 of King III states that the board of directors is the central governance and risk management authority. The board of directors can delegate these oversight functions to other management and governance committees. Therefore, management must support all areas of value creation and the optimum utilization of resources to achieve the company's objectives and to minimize or mitigate the inherent risks. Finally, first-line responsibility also includes compliance with legal, regulatory, and ethical guidelines and requirements (Eulerich 2021). Based on the above, the first hypothesis was formulated:

**H₁.** *Operational management compliance with legal, regulatory, and ethical requirements (MC) has a significant influence on governance procedures (GP).*

### 2.1.2. Second Line: Risk Management and Compliance Functions

The second LOD refers to the auxiliary functions that monitor risks continually in some way along with what actions management is taking. Management defines these functions to ensure that the first LOD is properly designed and functioning as required. The second line plays an important role in supporting the first line based on its experience in managing current or potential risks. Jobs in the second LOD vary by organization and industry, but typical functions in the second LOD include risk management, compliance, and quality. Risk management assist and monitors the implementation of practices defined by operational management and assists risk owners in identifying target risks and reporting risk-related information throughout the organization. The compliance function monitors various identified risks such as compliance with applicable laws and regulations, expected ethical behavior, internal control, information and technology, security, and sustainability (The Institute of Internal Auditors (IIA) (2020)). The quality function is responsible for ensuring the quality of services. The second line of defense should also ensure management hierarchy, particularly concerning risk perspective (Eulerich 2021). Based on the governance system, second-line management may be assigned to the board of directors or to a lower hierarchical level. Based on the King III rules, the audit committee is responsible for carrying out risk management, governance, and internal control tasks. The audit committee must also ensure that a common assurance model is applied to provide a coordinated approach to all assurance activities (PWC 2017).

**H₂.** *Risk management, compliance, and quality functions (RM) has a significant influence on governance procedures (GP).*

### 2.1.3. Third Line: Internal Audit

In light of agency theory, the board of directors and the audit committee assign the internal audit body to provide advice and reports to management regarding the organization's internal control (Schreurs and Marais 2015). In a business environment characterized by change, the internal audit function should be keen to meet the renewed needs of organizations to support and enhance their competitive position (Chambers and Odar 2015). The

internal audit profession has taken important steps toward implementing joint assurance in business organizations. The internal audit profession provides independent assurance of the effectiveness of risk management and plays an important role in evaluating the effectiveness of the first and second line concerning achieving the objectives of control and risk management. Internal auditing is defined as a department that is not involved in the organization's direct management functions, but provides assurance services that support and assist management in making decisions. An organization's internal auditing department has two primary tasks that no other LOD can carry out. The first is to submit reports to management after an assessment of all other lines of defense in the organization. The second task, which may be more difficult, is that it should cooperate and interact with external supervisory authorities, such as the external auditor, and meet his needs. A goal-oriented and structured internal audit process combined with expertise and knowledge help to achieve high-level audit results, thus supporting first- and second-line management, as well as supporting senior management and the supervisory board/audit committee (Christ et al. 2015; Carcello et al. 2018). Based on the above, the following hypothesis was formulated:

**H$_3$.** *The role of the assurance internal audit (IA) has a significant influence on governance procedures (GP).*

### 3. Previous Studies

The study conducted by Eulerich (2021) critically addressed the new model of the TLOD and discussed the similarities and differences with the old model. The study showed that the internal organization of governance functions, internal control system, risk management, compliance, and internal auditing still constitute a complex task. On the other hand, the new three-line model provides a high degree of flexibility and freedom in designing the management structure. This freedom means that among the many options available, the one that best reflects the specific characteristics of the company can be selected. For this reason, the new model should initially be seen as an additional aid that can support companies with integrated approaches in particular. The study carried out by Bantleon et al. (2021) aimed to analyze the determinants and challenges of implementing the TLOD among the various stakeholders in governance. In the study, 415 chief internal auditors from Austria, Germany, and Switzerland were surveyed to analyze the determinants that help implement the TLOD model without any difficulties, and to explore the extent of coordination between the internal audit function and governance stakeholders. The study results show great variance. If the company is listed, there are fewer coordination issues with the board of directors and external auditors. The results also indicate that a great fit with the international professional framework increases challenges with the compliance function, but reduces challenges with the external auditor. In addition, the results show significant variance in the extent of coordination challenges dependent on different determinants and the respective governance stakeholder. Bäßler and Eulerich (2022) implemented a framework that redefines the role of the internal auditor using predictive process monitoring within the three-line model. The study analyzed two publicly available event logs and proposed time- and state-based bucketing of prefixes in combination with a risk-based cost model for threshold optimization. In addition, the study used machine learning methods to predict the process outcomes. The study showed that by clustering traces according to their state and remaining time, internal auditors can use process predictions to provide assurance, reduce risk, and prevent undesired outcomes.

Nurdiani (2022) investigated the relationship between the elements of the TLOD model in different banks in Indonesia. The results showed that the TLOD model was able to not only measure the financial position of the bank, but also to reinforce the basic principles, expand the scope, and explain how the main roles cooperate within the bank to enhance corporate governance in terms of strength and risk management. According to the results, some banks have already reduced their risk by efficiently implementing the TLOD model.

Iskak and Muslih (2022) discussed the impact of the TLOD model on the performance of state-owned enterprises in Indonesia. The study sample included practitioners, observers, and academics. The results indicated that the first line of defense (LOD) has a positive impact on the company's performance, the second LOD has a negative impact on the company's performance, and the third defense risk (internal audit function) does not affect the company's performance. In addition, the audit committee amended the effect of the third line of defense (internal audit) on the company's performance, while it mitigated the impact of the second LOD (risk management unit) on the company's performance.

Luburic's (2017) study focused on the role of the TLOD model in enhancing the efficiency of managing operational risks (those that arise as a result of human factors and unsuccessful processes and systems, as well as those that can occur as a result of external events). The study confirmed that the TLOD can be strengthened through the synergy between the principles of risk management and the principles of quality management. The synergy and integration of quality management principles into the company's systems and processes will significantly enhance the TLOD in terms of the effective management of operational risks. Chambers and Odar (2015) concluded that the TLOD approach was not entirely effective and gave a false sense of reassurance. The internal audit function needs to move firmly into the area of corporate governance; to review corporate governance more effectively and to provide more reliable assurance to boards. A study (Davies and Zhivitskaya 2018) analyzed the TLOD in terms of whether it was a strong regulatory framework, or just lines in the sand. The study showed that regulators tried to strengthen governance mechanisms by applying the "three lines of defense" model to include risk management in all financial companies. The study found that this form is in use in several countries, but its origins are obscure, and its effectiveness has not been tested. It is not yet possible to make a final judgment on its effectiveness.

Several studies conducted by international audit firms (EY 2013; KPMG 2012; PWC 2017) confirmed that companies that suffer from weaknesses in applying the TLOD model may face a host of challenges: inconsistent reporting, gaps in risk coverage, and overwork.

## 4. Practical Method of Research

This section describes the questionnaire, the research sample, and the data collection and analysis.

### 4.1. Questionnaire Design

The questionnaire consisted of two parts (Appendix A). The first part contained scales that measure the demographic characteristics of the sample, while the second part contained axes that measure independent and dependent variables. Section 2 contained four axes: the first axis used six questions to measure the commitment of the operational management to legal, regulatory, and ethical requirements (MC); the second axis used nine questions to measure the commitment to risk management and compliance functions and quality (RM); the third axis used seven questions to measure the role of the internal assurance ascertainment according to the three lines of defense model (IA); and the fourth and final axis used six questions to measure the governance procedures (GP). The questionnaire was developed based on previous studies and the TLOD model issued by the IIA (Table 1). A five-point Likert scale was used to answer the questions. Finally, the data were analyzed using Smart PLS 3.3.3 software.

**Table 1.** Previous studies that were used in building the model and the questionnaire.

| Variables | Previous Studies |
| --- | --- |
| Operational management compliance with legal, regulatory, and ethical requirements (MC) | Laloux (2017); Pricewaterhouse Coopers (PWC) (2016); The Institute of Internal Auditors—Australia (2018); Eulerich (2021); The Institute of Internal Auditors (IIA) (2020); Roussy and Rodrigue (2018); |
| Risk management, compliance, and quality functions (RM) | Rittenberg (2013); Eulerich (2021); The Institute of Internal Auditors (IIA) (2020); Roussy and Rodrigue (2018); Luburic (2017) |
| The role of the assurance internal audit (IA) | Rossouw and Marais (2015); Sarens et al. (2012); The Institute of Internal Auditors—Australia (2018); Eulerich (2021); The Institute of Internal Auditors (IIA) (2020); Luburic (2017); Roussy and Rodrigue (2018) |
| Governance procedures (GP) | Roussy and Rodrigue (2018); Fanning and Piercey (2014); IFAC and IIA (2018) |

*4.2. Participants*

The study population consisted of internal auditors, financial managers and their assistants, general managers, board members, and members of the audit committee in industrial companies in the Sultanate of Oman, from which a sample was chosen. The questionnaire was distributed to the study sample electronically; 123 valid questionnaires were received for analysis. The response rate was medium because the questionnaire was distributed electronically to the sample and was followed up by the researchers.

Table 2 shows some demographic characteristics of the sample. The industrial sector was chosen because it is one of the most important sectors in the Omani economy after the oil sector. During 2021, the sector was one of the main sources of economic growth, effectively contributing to the increase in the country's exports. According to statistics issued by the National Centre for Statistics and Information, the non-oil industries sector recorded a growth of 5.7% during the year, bringing its contribution to the GDP to OMR 1.5 billion (about USD 4 billion).

**Table 2.** Demographic characteristics (*n* = 123).

| Qualification | No. | Major | No. | Experiences | No. |
| --- | --- | --- | --- | --- | --- |
| Diploma | 23 | Accounting | 57 | <5 years | 28 |
| Bachelor | 81 | Finance | 48 | 5–10 years | 51 |
| Postgraduate | 19 | Others | 18 | >10 years | 44 |

## 5. Data Analysis and Results

According to Henseler et al. (2009), the research analysis was based on a two-step way for reporting PLS-SEM findings using Smart PLS 3.3.3. The first step is the measurement model assessment, and the second step is the structural model assessment (Durrah and Kahwaji 2023). Operational management compliance with legal, regulatory, and ethical requirements, risk management, compliance, and quality functions, and the role of the assurance internal auditor were used as exogenous constructs, and governance procedures were used as endogenous constructs, as shown in Figure 1.

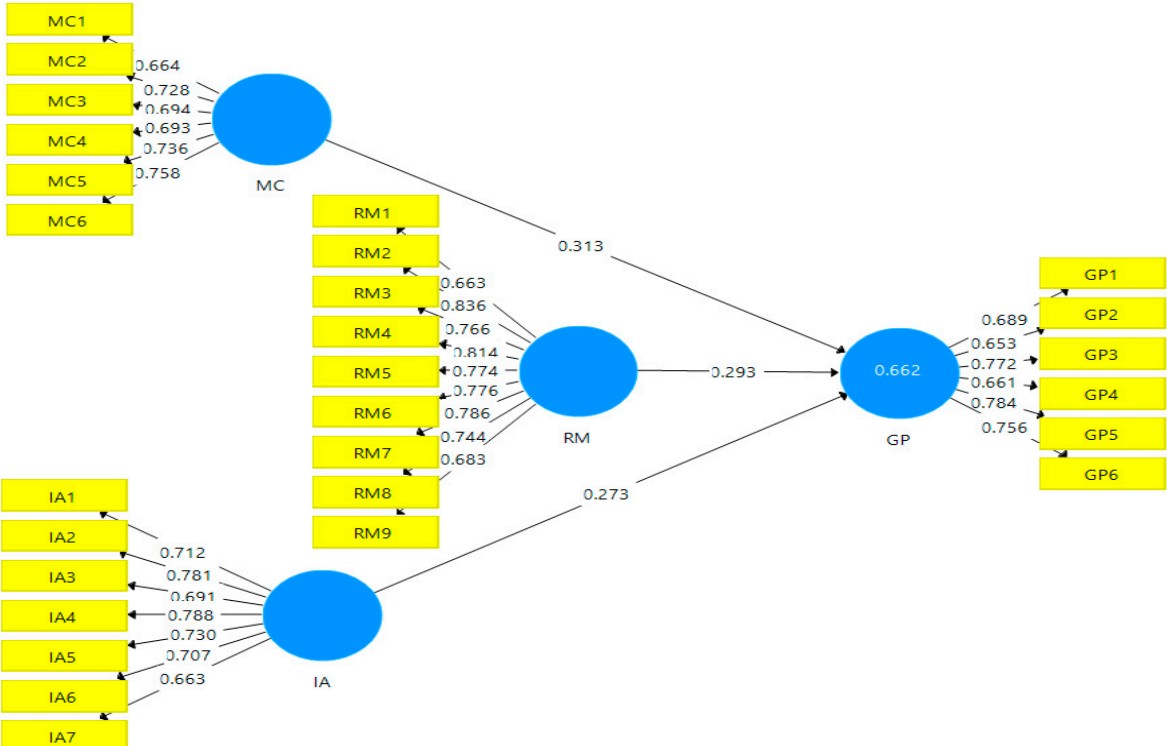

**Figure 1.** Structural model assessment.

### 5.1. First Step: Measurement Model Assessment

For the assessment of the measurement model, outer loading, Cronbach's alpha ($\alpha$), composite reliability (CR and rho_A), average extracted variance (AVE), and discriminant validity were examined, as shown in Figure 1 and Tables 3 and 4.

**Table 3.** Measurement model assessment.

| Construct | Outer Loading | Convergent Validity | | | |
|---|---|---|---|---|---|
| | | $\alpha$ | rho_A | CR | AVE |
| (MC) | ($\overline{x}$ = 4.06 and $\sigma$ = 0.546) | | | | |
| $MC_1$ | 0.664 | | | | |
| $MC_2$ | 0.728 | | | | |
| $MC_3$ | 0.694 | 0.806 | 0.806 | 0.861 | 0.508 |
| $MC_4$ | 0.693 | | | | |
| $MC_5$ | 0.736 | | | | |
| $MC_6$ | 0.758 | | | | |
| (RM) | ($\overline{x}$ = 4.02 and $\sigma$ = 0.615) | | | | |
| $RM_1$ | 0.663 | | | | |
| $RM_2$ | 0.836 | | | | |
| $RM_3$ | 0.766 | | | | |
| $RM_4$ | 0.814 | | | | |
| $RM_5$ | 0.774 | 0.911 | 0.909 | 0.925 | 0.581 |
| $RM_6$ | 0.776 | | | | |
| $RM_7$ | 0.786 | | | | |
| $RM_8$ | 0.744 | | | | |
| $RM_9$ | 0.683 | | | | |

**Table 3.** *Cont.*

| Construct | Outer Loading | Convergent Validity | | | |
|---|---|---|---|---|---|
| | | $\alpha$ | rho_A | CR | AVE |
| (IA) | ($\bar{x}$ = 4.07 and $\sigma$ = 0.562) | | | | |
| IA$_1$ | 0.712 | | | | |
| IA$_2$ | 0.781 | | | | |
| IA$_3$ | 0.691 | | | | |
| IA$_4$ | 0.788 | 0.850 | 0.853 | 0.886 | 0.527 |
| IA$_5$ | 0.730 | | | | |
| IA$_6$ | 0.707 | | | | |
| IA$_7$ | 0.663 | | | | |
| (GB) | ($\bar{x}$ = 3.98 and $\sigma$ = 0.579) | | | | |
| GB$_1$ | 0.689 | | | | |
| GB$_2$ | 0.653 | | | | |
| GB3 | 0.772 | | | | |
| GB$_4$ | 0.661 | 0.814 | 0.815 | 0.866 | 0.520 |
| GB$_5$ | 0.784 | | | | |
| GB$_6$ | 0.756 | | | | |

Note: $\alpha$ = Cronbach's alpha; CR and rho_A = composite reliability, AVE = average variance extracted.

**Table 4.** Discriminant validity and multicollinearity.

| Construct | MC | RM | IA | GB |
|---|---|---|---|---|
| MC | **0.713** | | | |
| RM | 0.694 | **0.762** | | |
| IA | 0.703 | 0.660 | **0.726** | |
| GP | 0.665 | 0.649 | 0.647 | **0.721** |

The values in boldface are the square root of AVE.

It is evident from Figure 1 and Table 3 that the outer loading values for all study constructs were greater than 0.6 (Hair et al. 2010). The Cronbach's alpha ($\alpha$) values were more than 0.6 (Tawfik et al. 2022; Tawfik and Durrah 2023), and the composite reliability values (rho_A and CR) were higher than 0.7 (Dijkstra and Henseler 2015; Raykov 1997). Moreover, the results of the study showed that the average extracted variance values (AVE) exceeded the 0.50 cut-off (Hair and Lukas 2014). So, it can be said that the convergent validity is met. On the other hand, Table 3 shows that the means of the research variables were high, with a value ranging between 3.98 and 4.07, and that the standard deviation values were of low dispersion.

Fornell and Larcker's (1981) criterion was implemented to determine the discriminant validity, which compares the correlations between the square root of the AVE and the constructs. The results in Table 4 point out that all constructs (MC, RM, IA, and GB) had values (in boldface) greater than the other construct correlation values. Thus, these findings confirm adequate discriminant validity (Durrah 2022; Gye-Soo 2016).

### 5.2. Second Step: Structural Model Assessment

To determine the direct impact of the independent variables (MC, RM, and IA) on the dependent variable (GB), structural equation modeling (SEM) was implemented using Smart PLS. To estimate the path coefficients' significance, bootstrapping was performed through Smart PLS, as shown in Table 4, showing that all of the path coefficient absolute values were more than 0.1, indicating the effect of the independent variable predictor on the dependent variable (Nasaruddin et al. 2018).

From the findings in Table 5, it can be observed that the TLOD (the operational management's commitment to regulatory and ethical legal requirements (MC), risk management

and compliance functions (RM), and the role of assurance internal auditor (IA)) had an effect on strengthening CG procedures (GP).

**Table 5.** Structural model assessment results.

| Hypothesis | Path Coefficient | T-Statistic | *p*-Value | Decision | $f^2$ | $R^2$ | $Q^2_{predict}$ | GoF |
|---|---|---|---|---|---|---|---|---|
| H$_1$: (MC → GB) | 0.313 | 2.889 | 0.004 | Supported ** | 0.080 | | | |
| H$_2$: (RM → GB) | 0.293 | 2.413 | 0.016 | Supported * | 0.083 | 0.662 | 0.327 | 0.197 |
| H$_3$: (IA → GB) | 0.273 | 2.544 | 0.011 | Supported * | 0.069 | | | |

Significant at $p$ * < 0.05, $p$ ** < 0.01.

MC had a positive influence on GP at the significance level of 0.01 (T-Statistic = 2.889, *p*-value = 0.004), and RM had a positive influence on GP at the significance level of 0.05 (T-Statistic = 2.413, *p*-value = 0.016). Also, IA had a positive influence on GP at the significance level of 0.05 (T-Statistic = 2.544, *p*-value = 0.011).

Regarding the effect size $f^2$, H$_1$, H$_2$, and H$_3$ had a small effect (0.080, 0.083, and 0.069, respectively) according to Cohen (1988). Thus, H$_1$, H$_2$, and H$_3$ are supported by the results. Moreover, the determination coefficient ($R^2 = 0.662$) indicates that there is a moderate interpretive ability, as explained by Falk and Miller (1992).

In addition, the predictive capacity of the model was analyzed to interpret the $Q^2_{predict}$ values in this study. The predictive relevance of the TLOD (MC, RM, and IA) was higher than zero (0.327), supporting the claim that this study model has adequate ability to predict (Fornell and Cha 1994; Hair et al. 2019). Furthermore, the model fit value was GoF = 0.594. Thus, we can conclude that the GoF model is highly adequate for considering model viability (Durrah et al. 2022; Wetzels et al. 2009).

## 6. Discussion

The TLOD model has become a trusted tool in a wide range of industries, addressing issues related to governance, risk management, and control. The IIA's latest update in 2020 changed the way organizations examine risk, controls, accountability, and assurance. Only a few empirical studies have addressed the use of the TLOD model. The operational management's commitment to legal, regulatory, and ethical requirements plays an important role in the strengthening of governance. The organization's operational management must have the responsibility and accountability for assessing, controlling, and mitigating risks. Lines of defense play an important and effective role in strengthening the communication between risk management and control, and it was considered by the Basel Committee for Banking Regulation as the best method for banking control. Leech and Hanlon (2016) indicate that operational management is responsible for maintaining effective internal controls and implementing risk and control procedures daily. Table 4 shows the existence of positive relationships between the TLOD (operational management compliance with legal, regulatory, and ethical requirements, risk management, compliance, and quality functions, and the role of the assurance internal auditor were used as exogenous constructs) and the strengthening of governance. The results of the statistical analysis showed that there is a significant relationship between the operational management's commitment to legal, regulatory, and ethical requirements and the strengthening of governance in economic units. These results are consistent with the results of previous studies conducted in several countries (Leech and Hanlon 2016; Decaux and Sarens 2015; EY 2013). The reason for this, according to the opinion of the respondents, is that the first LOD provides internal control mechanisms to follow up the daily work and take appropriate corrective measures when deviations or errors occur, in addition to making sure that the control procedures work effectively.

The second LOD sets policy and directives for risk management, provides risk advice and guidance, and monitors the first LOD on effective risk management. According to The Institute of Internal Auditors (IIA) (2020), the second LOD helps develop and

implement risk management practices and works continuously to develop them, in addition to achieving risk management objectives such as compliance with laws, instructions, control methods, and information security. The results of the statistical analysis showed that there is a significant relationship between risk management, compliance functions, quality, and the enhancement of governance. These results are consistent with the results of previous studies (KPMG 2012; PWC 2017; Luburic 2017). The reason for this is that the second line supports the management of the first line in the mitigation of all risks arising in the first line. In general, the second line can be viewed as a specialized function that supports the first line. Interestingly, all first- and second-line functions can be combined or separated.

The third variable in the model is internal audit. Internal auditing plays a pivotal role in the audit process and contributes significantly to strengthening CG. The results of the statistical analysis showed that there is a statistically significant relationship between internal auditing and the enhancement of governance in economic units. The reason for this, according to the respondents, is that internal auditing evaluates the efficiency and effectiveness of the first and second lines of defense, in addition to ensuring the effectiveness of governance and risk management. These findings are consistent with the findings of previous studies (Anderson and Eubanks (2015); Eulerich et al. (2015); Lewis (2014); King III report). The internal audit function plays an important role in strengthening control procedures and constitutes an essential element in supporting joint assurance. In addition, Harrington and Piper (2015) found that 54% to 64% of auditors globally believe that internal auditing is an independent function in their organizations, and that it is responsible for strengthening governance procedures.

## 7. Implications

The results of this study could have significant implications for the industrial companies in the Sultanate of Oman, because the study of the TLOD model and its impact on enhancing governance procedures is very beneficial in several aspects. First, the study provides a better understanding of the conditions under which the TLOD can operate successfully. In light of the development of the business environment, it has become necessary to search for new control methods to enhance the credibility of the information contained in these reports. In addition, the increasing level of complexity in the business environment in general requires an improvement in the efficiency and effectiveness of internal control mechanisms to provide assurances regarding the ability of organizations to perform their work. Second, in light of the results of this study, companies can coordinate the control of internal functions in order to avoid the duplication of efforts and gaps in risk coverage. The results of the study also help corporate management to coordinate assurance activities and work methods between different functions to increase assurance effectiveness, as well as ensuring continuous improvement.

The different perspectives on the TLOD model covered in this study form a management strategy that helps companies improve control procedures in light of the available economic resources. This strategy also helps managers to choose and invest in the most appropriate dimension of TLOD to significantly improve their performance (Laloux 2017; Pricewaterhouse Coopers (PWC) (2016)). Hence, a comprehensive and multifaceted visualization of the adoption of the TLOD can help managers to identify the available options (The Institute of Internal Auditors—Australia 2018).

Despite all of the above contributions, the study has several limitations. These limitations may be addressed in future studies that may investigate the relationship between the elements highlighted in this research as well as other related fields. First, the research tool was a survey questionnaire based on the opinions of a group of workers in industrial companies that may be related to auditing and control. In this regard, there may be some bias in their opinions, even if the research tool has already been tested for correctness or reliability. This bias can be mitigated if the opinions of external parties such as allied partners, customers, competitors, and suppliers are considered. Furthermore, it would be useful to evaluate the annual reports to validate the information provided by the respon-

dents. Second, all key elements were collected and measured only once during one period, so it is important to consider the long-term effects, especially those that may affect the development and establishment of the services of the internal audit function along with the presence of senior management support. Third, the data were collected from one country (Oman), so potential cultural differences, particularly differences between developed and developing societies affecting performance practices, must be considered. To generalize the ideas and concepts of the review, the research framework needs further research and needs to be attached to samples from different countries. In addition, current cultural differences can influence individuals' opinions about certain key activities, so future research can test more hypotheses.

## 8. Conclusions

The TLOD model is a simple and effective way to enhance communication about risk management and control by clarifying the essential roles and duties of each of the three lines. Lines of defense reduce information asymmetry between principles and agents at all levels of the hierarchy and reduce the risk of discretionary decisions by agents. It is important to note that the IIA paper notes that senior management and boards of directors act as stewards of the TLOD, and are not active participants or additional lines of defense. This model achieves effective results if the three lines of the structure act as organized lines of defense. A questionnaire was developed and distributed to a sample of internal auditors, financial managers and their assistants, general managers, board members, and members of the audit committee in a sample of industrial companies in the Sultanate of Oman. The results indicated that there is a significant impact of the three axes (the operational management's commitment to legal, regulatory, and ethical requirements, risk management and compliance and quality functions, and the role of the assurance and advisory internal auditor according to the three LOD model) on strengthening governance. The results also showed a significant correlation between the TLOD and the effectiveness of governance in non-oil industrial companies in the Sultanate of Oman. The results of this study can have significant effects on senior management to reveal the risks related to the policies, processes, and structures of organizational governance and to recommend to the board of directors improvements in the methods used to manage the risks associated with the policies and operations of the company. In addition, the internal control framework can be improved by relying on international frameworks and standards for control. On the other hand, companies with internal control problems can plan to use the TLOD model.

Future research can address the external auditor's dependence on the TLOD model in developing a form to periodically examine the roles of the internal auditor, senior management, the board of directors, and affiliated committees. Future research can also address the evaluation of the CG structure and the impact of the organizational structure and culture on the overall control environment and risk management strategy. Our results help firms to determine whether or not there is a particular challenge, and which specific factor may be the most influential.

**Author Contributions:** Conceptualization, O.I.T. and K.A.A.; methodology, O.I.T. and K.A.A.; software, O.D.; formal analysis, O.D.; investigation, O.D.; writing—original draft preparation, O.I.T.; writing—review and editing, O.I.T. and O.D.; project administration, O.I.T. All authors have read and agreed to the published version of the manuscript.

**Funding:** This research received no external funding.

**Data Availability Statement:** Not applicable.

**Conflicts of Interest:** The authors declare no conflict of interest.

**Appendix A  The Questionnaire**

**Demographic characteristics**

**Qualification**

Diploma          Bachelor          Postgraduate

**Major**

Accounting          Finance          Others

**Experiences**

<5 years          5–10 years          >10 years

| No. | Questions | The scale | | | | |
|---|---|---|---|---|---|---|
| | | Strongly agree | agree | neutral | disagree | Strongly disagree |
| First axis | The administration is committed to the following procedures | | | | | |
| 1 | Determine the regulatory requirements for the use of economic resources | | | | | |
| 2 | Adhere to local legislation when implementing various activities | | | | | |
| 3 | Adhere to ethical rules when making various decisions | | | | | |
| 4 | Reviewing the internal control procedures that are put in place within the sub-departments | | | | | |
| 5 | The administration determines the methods of communication between it and the Board of Directors | | | | | |
| 6 | The administration is committed to the rules of ethical behaviour in terms of integrity and reviewing the management's minutes and decisions | | | | | |
| Second axis | The Board of Directors, through its committees, is committed to: | | | | | |
| 1 | Reviewing the oversight and supervision reports submitted by the subsidiary departments of the company | | | | | |
| 2 | Checking contracts with third parties and their compliance with laws | | | | | |
| 3 | Determine the level of risk that is accepted in the business | | | | | |
| 4 | Monitor the extent of compliance with the laws and legislation regulating work | | | | | |
| 5 | Verifying information security policies within the company | | | | | |
| 6 | Develop and review the company's quality policies | | | | | |
| 7 | Reviewing change policies and accompanying control procedures | | | | | |
| 8 | Placing qualified cadres in sub-units | | | | | |
| 9 | Setting rules of ethical behavior in terms of integrity and reviewing plans and management decisions | | | | | |
| Third axis | The company's internal audit staff has | | | | | |
| 1 | Provide information to management and communicate with the external auditor | | | | | |
| 2 | Full independence in his work to carry out his assurance services | | | | | |
| 3 | The Internal Audit Authority has qualified and sufficient staff to carry out its work and provide its services | | | | | |
| 4 | Active participation in determining the effectiveness of governance structures and risk management policies | | | | | |

| 5 | The advice of the internal auditor is sought when important decisions are taken by the Board of Directors | | | | | |
| 6 | Submitting annual reports showing the results of the internal audit work to the Board of Directors | | | | | |
| 7 | Periodic reports are submitted in case the auditor encounters obstacles that affect his independence | | | | | |
| Fourth axis | Governance procedures are described in the company | | | | | |
| 1 | Existence of a written and effective framework that clarifies the governance procedures within the company | | | | | |
| 2 | The procedures applied by the company guarantee the rights of all stakeholders fairly | | | | | |
| 3 | Disclosure of all important information to stakeholders | | | | | |
| 4 | The duties of management, the board of directors and the internal auditor are clear and non-overlapping | | | | | |
| 5 | The goals are strategic for the company and the set of values and principles are known to all. | | | | | |
| 6 | The company's liability policies are clear | | | | | |

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
