# Peer review of "The Role of The Internal Auditor in Strengthening the Governance of Economic Organizations Using the Three Lines of Defense Model"

_jrfm, doi:10.3390/jrfm16070341_

Round 1
Reviewer 1 Report
Article Review: The Role of The Internal Auditor in Strengthening the Governance of Economic Organizations Using the Three Lines of Defense Model
I would like to commend the author for the clear structure of the article and the organization of the information presented. The article's structure facilitates reader understanding and allows for a smooth reading experience. Additionally, the clarity of the information shown is a strong point of the work.
Another positive aspect to highlight is the use of current literature in the article. The author demonstrates having conducted a comprehensive literature review and references relevant studies in the field. This strengthens the theoretical foundation of the work and contributes to the credibility of the presented conclusions.
However, I suggest that the author make a few revisions to specific points in the article.
Firstly, the abstract needs improvement. The study's objective should be clearly aligned with the objective presented in the introduction. Additionally, the author needs to highlight the relevance and contribution of the research explicitly. This will help the reader understand why the study is important and its main contributions.
Regarding the introduction, a research gap must be presented based on the literature review. I suggest moving the last lines of the introduction, which refer to the data collected from the questionnaire derived from the three lines of defense model and previous studies, to after the research question is presented. Furthermore, I recommend that these lines be rewritten more concisely and clearly, avoiding long sentences and using periods every one and a half pages, two pages written. This will aid in the comprehension of the text.
Another point to consider is the presentation of the article's contribution. In the introduction, the author should present the expected contribution of the study rather than the actual contribution. The contributions should be further developed in the implications and conclusions of the article.
These revision suggestions will strengthen the article, making it more cohesive and enhancing reader comprehension. Overall, the work has a solid and promising foundation, and with the suggested revisions, it will become an even more impactful article.
Author Response
Firstly, the abstract needs improvement. The study's objective should be clearly aligned with the objective presented in the introduction. Additionally, the author needs to highlight the relevance and contribution of the research explicitly. This will help the reader understand why the study is important and its main contributions. done p. 1
Regarding the introduction, a research gap must be presented based on the literature review. I suggest moving the last lines of the introduction, which refer to the data collected from the questionnaire derived from the three lines of defense model and previous studies, to after the research question is presented. Furthermore, I recommend that these lines be rewritten more concisely and clearly, avoiding long sentences and using periods every one and a half pages, two pages written. This will aid in the comprehension of the text. Done P.1-2
Another point to consider is the presentation of the article's contribution. In the introduction, the author should present the expected contribution of the study rather than the actual contribution. The contributions should be further developed in the implications and conclusions of the article. Done
Reviewer 2 Report
The paper aims to enhance corporate governance by applying the three lines of defense model to a sample of industrial companies in the Sultanate of Oman to diagnose the degree of companies' commitment to those lines and to diagnose weaknesses in their procedures. The paper is sound and I believe it will contribute to the literature, however I have the following concerns:
- Although the paper is still readable, there are many typos that must be corrected. just a few example: fefense model (page 2), The first line of defense 's required (page 2), defance (page 4), Knig III (page 4) and so forth.
- I do not understand the phrase "second jaffa line" on page 4.
- The introduction does mention policy implications derived from the results of the study, but the actual results themselves are not explained in the provided text. The introduction only alludes to the research objective of examining the role of the Three Lines of Defense Model in enhancing corporate governance in industrial companies in the Sultanate of Oman. In addition, the introduction briefly mentions that data were collected from a questionnaire derived from the Three Lines of Defense Model and previous studies, however, just at the end of the section. I recommend the authors to revise and rewrite this section following the research question: approaches, brief findings and contributions. I urge removing implications from this section, as it already appears in section 6.
- There are inapporiate use of question marks in the manuscript such as "that management takes to achieve the company's goals and efficiently utilize resources?", analyzed the defense lines model in terms of whether it is a strong regulatory framework, or just lines in the sand?.
- The use of "you" should be reconsidered in the following: "You may face a range of challenges: inconsistent reporting, gaps in risk coverage, overwork"
- The authors articulate that "only a few empirical studies have addressed the use of Three Lines of Defense model in Oman". Please kindliy include some of these studies in the literature review section.
- the sentence "Second: In light of the results of this study, companies can coordinate control functions" is incomplete.
Author Response
However I have the following concerns:
- Although the paper is still readable, there are many typos that must be corrected. just a few example: fefense model (page 2), The first line of defense 's required (page 2), defance (page 4), Knig III (page 4) and so forth. Done I correct all.
- I do not understand the phrase "second jaffa line" on page 4. done I correct it
- The introduction does mention policy implications derived from the results of the study, but the actual results themselves are not explained in the provided text. The introduction only alludes to the research objective of examining the role of the Three Lines of Defense Model in enhancing corporate governance in industrial companies in the Sultanate of Oman. In addition, the introduction briefly mentions that data were collected from a questionnaire derived from the Three Lines of Defense Model and previous studies, however, just at the end of the section. I recommend the authors to revise and rewrite this section following the research question: approaches, brief findings and contributions. I urge removing implications from this section, as it already appears in section 6. done p. 1-2
- There are inapporiate use of question marks in the manuscript such as "that management takes to achieve the company's goals and efficiently utilize resources?", analyzed the defense lines model in terms of whether it is a strong regulatory framework, or just lines in the sand?. Done
- The use of "you" should be reconsidered in the following: "You may face a range of challenges: inconsistent reporting, gaps in risk coverage, overwork". Done
- The authors articulate that "only a few empirical studies have addressed the use of Three Lines of Defense model in Oman". Please kindliy include some of these studies in the literature review section. Done (I mean only a few empirical studies have addressed the use of Three Lines of Defense model)
- the sentence "Second: In light of the results of this study, companies can coordinate control functions" is incomplete. Done I correct it.
Round 2
Reviewer 2 Report
The manuscript has been revised in line with the recommendations and I have no further considerations. However, the following issue seems not corrected in the revised version. I urge to fix this very minor problem.
"I do not understand the phrase "second jaffa line" on page 4. done I correct it"
Author Response
I do not understand the phrase "second jaffa line" on page 4. Done I correct it"